# WAFT: WARPING-ALONE FIELD TRANSFORMS FOR OPTICAL FLOW

**Yihan Wang & Jia Deng**
Department of Computer Science
Princeton University
{yw7685, jiadeng}@princeton.edu

## ABSTRACT

We introduce Warping-Alone Field Transforms (WAFT), a simple and effective method for optical flow. WAFT is similar to RAFT but replaces cost volume with high-resolution warping, achieving better accuracy with lower memory cost. This design challenges the conventional wisdom that constructing cost volumes is necessary for strong performance. WAFT is a simple and flexible meta-architecture with minimal inductive biases and reliance on custom designs. Compared with existing methods, WAFT ranks 1st on Spring, Sintel, and KITTI benchmarks, achieves the best zero-shot generalization on KITTI, while being $1.3 - 4.1\times$ faster than existing methods that have competitive accuracy (e.g., $1.3\times$ than Flowformer++, $4.1\times$ than CCMR+). Code and model weights are available at https://github.com/princeton-vl/WAFT.

## 1 INTRODUCTION

Optical flow is a fundamental low-level vision task that estimates per-pixel 2D motion between video frames. It has many downstream applications, including 3D reconstruction and synthesis (Ma et al., 2022; Zuo & Deng, 2022), action recognition (Sun et al., 2018b; Piergiovanni & Ryoo, 2019; Zhao et al., 2020b), frame interpolation (Xu et al., 2019; Liu et al., 2020; Huang et al., 2020), and autonomous driving (Geiger et al., 2013; Menze & Geiger, 2015; Janai et al., 2020).

Cost volumes (Sun et al., 2018a; Ilg et al., 2017) with iterative updates (Teed & Deng, 2020; Wang et al., 2024) has become a standard design in most state-of-the-art methods (Sun et al., 2018a; Dosovitskiy et al., 2015; Xu et al., 2017; Teed & Deng, 2020; Huang et al., 2022; Wang et al., 2024; Morimitsu et al., 2025), especially when both accuracy and efficiency are taken into account. Previous work (Sun et al., 2018a; Teed & Deng, 2020) regards cost volumes as a more effective representation than image features, as it explicitly models the visual similarity between pixels.

However, constructing cost volumes is expensive in both time and memory (Zhao et al., 2024; Xu et al., 2023a). The cost increases quadratically with the radius of the neighborhood. As a result, cost volumes are often constructed from low resolution features, limiting the ability of the model to handle high-resolution input images.

In this paper, we challenge the conventional wisdom that cost volume is necessary for strong performance with high efficiency, and introduce Warping-Alone Field Transforms (WAFT), a simplified design that replaces cost volumes with warping and achieves state-of-the-art accuracy across various benchmarks with high efficiency.

For each pixel in frame 1, instead of computing its similarities against many pixels in frame 2, warping simply fetches the feature vector of the corresponding pixel given by the current flow estimate; this enables memory-efficient high-resolution processing and leads to better accuracy.

The design of WAFT is simple, with flow-specific designs kept to the minimum. WAFT consists of an input encoder that extracts features from individual input frames and a recurrent update module that iteratively updates flow. Compared to other RAFT-like architectures, WAFT is much simplified because it does not use cost volumes and has removed the context encoder that provides extra features for the update module. WAFT is designed to function as a meta-architecture for optical flow in

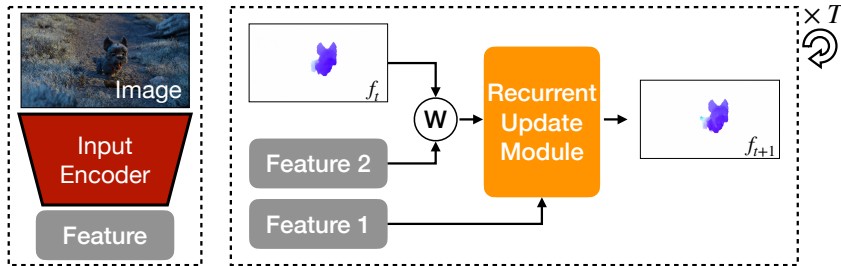

Figure 1: The meta-architecture of WAFT consists of an input encoder and a recurrent update module. We first extract image features from the input encoder, and then use these features to iteratively update the flow estimate for $T$ steps. At each step, we perform feature indexing through a lightweight backward warping on the feature of frame 2, removing the dependency on expensive cost volume used by previous work.

the sense that the individual components including input encoder and the update unit do not require custom designs and can use existing off-the-shelf (pretrained) architectures. In our experiments, we evaluate different choices (such as ResNet (He et al., 2016) and DPT (Ranftl et al., 2021)) that have different accuracy-efficiency trade-offs.

WAFT achieves state-of-the-art performance across various benchmarks with high efficiency and a simple design. Using a Twins (Chu et al., 2021) backbone pre-trained only on ImageNet, WAFT ranks first on Spring, second on KITTI, and is competitive on Sintel. It also achieves the best zero-shot cross-dataset generalization on KITTI. Using a stronger backbone, depth-pretrained DAv2 (Yang et al., 2024), WAFT outperforms existing methods on all public benchmarks. We achieve this with standard network architectures for the sub-modules (Dosovitskiy et al., 2020; He et al., 2016; Ranftl et al., 2021), removing custom designs typically needed in prior work, while being $1.3 - 4.1\times$ faster than existing methods that have competitive accuracy (e.g., $1.3\times$ than Flow-former++ Shi et al. (2023), $4.1\times$ than CCMR+ Jahedi et al. (2024)).

Our main contributions are two-fold: (1) we challenge the conventional wisdom that cost volume is a key component for achieving state-of-the-art accuracy and efficiency for optical flow; (2) we introduce WAFT, a warping-based meta-architecture that is simpler and achieves state-of-the-art accuracy with high efficiency.

## 2 RELATED WORK

**Estimating Optical Flow**  Traditional methods treated optical flow as a global optimization problem that maximizes visual similarity between corresponding pixels (Horn & Schunck, 1981; Zach et al., 2007; Chen & Koltun, 2016; Brox et al., 2004). These methods apply coarse-to-fine warping (Brox et al., 2004; Black & Anandan, 1996; Memin & Perez, 1998), a strategy theoretically justified by Brox *et al.* (Brox et al., 2004), to solve this optimization.

Today, this field is dominated by deep learning methods (Ilg et al., 2017; Dosovitskiy et al., 2015; Sun et al., 2018a; Zhao et al., 2020a; Hui et al., 2018; Teed & Deng, 2020; Sui et al., 2022; Sun et al., 2022; Deng et al., 2023; Huang et al., 2022; Shi et al., 2023; Weinzaepfel et al., 2022; 2023; Xu et al., 2022; 2023b; Leroy et al., 2023; Saxena et al., 2024; Jahedi et al., 2024; 2023; Luo et al., 2022; Zheng et al., 2022; Zhao et al., 2022; Luo et al., 2023; Jung et al., 2023; Luo et al., 2024; Zhou et al., 2024; Morimitsu et al., 2025), which can be categorized into two paradigms: direct or iterative. Direct methods (Dosovitskiy et al., 2015; Weinzaepfel et al., 2022; 2023; Saxena et al., 2024; Leroy et al., 2023; Xu et al., 2022) treat flow estimation as a standard dense prediction task (*e.g.* monocular depth estimation) and directly regress the dense flow field from large-scale pre-trained models. Iterative methods (Teed & Deng, 2020; Wang et al., 2024; Luo et al., 2024; Morimitsu et al., 2025; Zhou et al., 2024; Sun et al., 2018a; Huang et al., 2022) align more closely with traditional warping-based approaches, refining the flow predictions progressively. Most state-of-the-art methods (Teed & Deng, 2020; Wang et al., 2024; Huang et al., 2022; Shi et al., 2023; Luo et al., 2024; Morimitsu et al., 2025) follow the iterative paradigm due to its significantly higher efficiency than the direct ones.

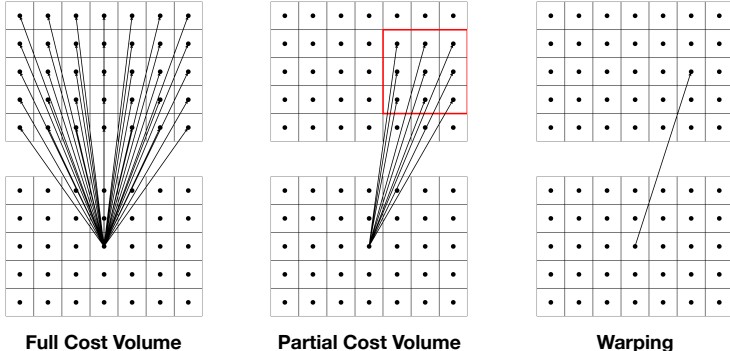

**Full Cost Volume**    **Partial Cost Volume**    **Warping**

Figure 2: For each pixel, the full cost volume calculates its visual similarity to all pixels in the other frame through correlation. The partial cost volume restricts the search range to the neighborhood of the corresponding pixel, marked by a red box. Compared with them, warping only uses the information from the corresponding pixel, offering better time and memory efficiency. This efficiency enables high-resolution processing, which leads to improved accuracy.

Cost volumes (Sun et al., 2018a; Dosovitskiy et al., 2015) have been regarded as a standard design in iterative methods (Teed & Deng, 2020; Wang et al., 2024; Morimitsu et al., 2025; Huang et al., 2022; Shi et al., 2023; Xu et al., 2023b; Luo et al., 2024). Prior work (Teed & Deng, 2020; Sun et al., 2018a) empirically shows the effectiveness of cost volumes in handling large displacements. Many iterative methods (Teed & Deng, 2020; Wang et al., 2024; Morimitsu et al., 2025) adopt partial cost volumes (Sun et al., 2018a) to avoid the quadratic computational complexity of full 4D cost volumes; they restrict the search range of each pixel in frame 1 to the neighborhood of its corresponding pixel in frame 2. However, they still suffer from the high memory consumption inherent in cost volumes (Xu et al., 2023a; Zhao et al., 2024).

WAFT is a warping-based iterative method. We achieve state-of-the-art performance across various benchmarks without constructing cost volumes, challenging the conventional wisdom established by previous work (Sun et al., 2018a; Teed & Deng, 2020; Huang et al., 2022). Warping no longer suffers from high memory consumption inherent in cost volumes, which enables high-resolution indexing and therefore improves accuracy.

**Vision Transformers** Vision transformers (Dosovitskiy et al., 2020) have achieved significant progress across a wide range of visual tasks (Yang et al., 2024; Kirillov et al., 2023; Oquab et al., 2023; He et al., 2022; Rombach et al., 2022). In the context of optical flow, most direct methods (Weinzaepfel et al., 2022; 2023; Saxena et al., 2024) regress flow from a large-scale pre-trained vision transformer with a lightweight flow head. Many iterative methods (Huang et al., 2022; Shi et al., 2023; Luo et al., 2024; Zhou et al., 2024) design task-specific transformer blocks to process cost volumes.

WAFT adopts similar designs to DPT (Ranftl et al., 2021) in its recurrent update module, which implicitly handles large displacements in optical flow through the transformer architecture. We empirically show that this is crucial to make warping work. WAFT can also benefit from large-scale pre-trained transformers like existing methods (Saxena et al., 2024; Weinzaepfel et al., 2022; 2023; Zhou et al., 2024), with minimal additional flow-specific designs.

## 3 BACKGROUND

In this section, we first review current cost-volume-based iterative methods and discuss the drawbacks of cost volumes. Then we introduce warping and compare it to cost volumes.

### 3.1 ITERATIVE METHODS WITH COST VOLUME

Given two adjacent RGB frames, optical flow predicts pixel-wise 2D motion between adjacent frames. Current iterative methods (Teed & Deng, 2020; Huang et al., 2022; Wang et al., 2024)

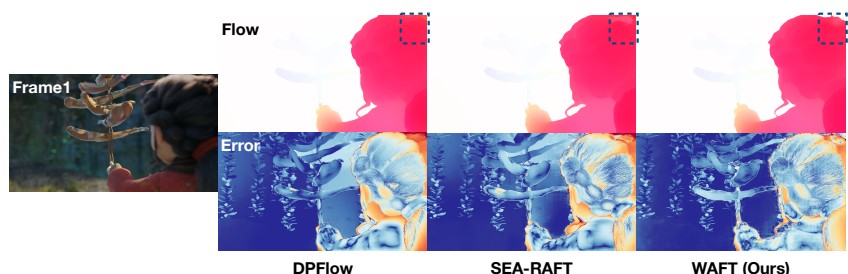

Figure 3: Visualizations of different methods on Spring (Mehl et al., 2023). WAFT, benefiting from high-resolution indexing, obtains sharper boundaries and lower errors than low-resolution approaches.

consist of two parts: (1) input encoders that extract dense image features at low resolution, and (2) a recurrent update module that iteratively refines the flow estimate.

Denoting the two frames as $I_1, I_2 \in \mathbb{R}^{H \times W \times 3}$, the input encoder $F$ maps $I_1, I_2$ to low-resolution dense features $F(I_1), F(I_2) \in \mathbb{R}^{h \times w \times d}$, respectively. A cost volume $V \in \mathbb{R}^{h \times w \times h \times w}$ is built on these features, which explicitly models the correlation between pixels ($p \in I_1, p' \in I_2$):

$$V_{p,p'} = F(I_1)_p \cdot F(I_2)_{p'}$$

where $\cdot$ represents the dot product of two vectors. At each step, the recurrent update module indexes into the cost volume using the current flow estimate and predicts the residual flow update. Several methods (Huang et al., 2022; Shi et al., 2023) directly process $V$ for better performance, but tend to be more costly.

Partial cost volume (Sun et al., 2018a) is introduced to reduce the cost by avoiding the construction of a full 4D cost volume. Given the current flow estimate $f_{cur} \in \mathbb{R}^{h \times w \times 2}$ and a pre-defined look-up radius $r$, partial cost volume $V_{par} : \mathbb{R}^{h \times w \times 2} \rightarrow \mathbb{R}^{h \times w \times (r^2)}$ implements an on-the-fly partial construction by restricting the indexing range of a pixel $p \in I_1$ to the neighborhood of its corresponding pixel $p + (f_{cur})_p \in I_2$, formulated as:

$$V_{par}(f_{cur}; r)_p = \text{concat}(\{V_{p,p'} | \forall p' \in I_2, \text{s.t.} \| p + (f_{cur})_p - p' \|_\infty \leq r \|)$$

where the operator "concat" concatenates all values inside the set into a vector. In practice, partial cost volumes are usually constructed at multiple scales (Teed & Deng, 2020; Wang et al., 2024) to improve the prediction of large displacements. Current methods also introduce context encoder (Sun et al., 2018a; Teed & Deng, 2020; Huang et al., 2022; Wang et al., 2024) to enhance the effectiveness of iterative refinement.

## 3.2 DRAWBACKS OF COST VOLUMES

**High Memory Cost** The main drawback of cost volume is its high memory consumption. Full or partial cost volume at high resolution are very expensive and often infeasible (Zhao et al., 2024; Xu et al., 2023a). Therefore, most iterative methods build the cost volume and index into it at 1/8 resolution. To further demonstrate the problem, we implement several variants of SEA-RAFT (Wang et al., 2024) that build partial cost volumes at different resolutions. We set the base channel dimension as 32, 64, and

| Method | Training Memory Cost (GiB) | | |
|---|---|---|---|
| | 1/8 Reso. | 1/4 Reso. | 1/2 Reso. |
| SEA-RAFT | 14.1 | 25.8 | OOM |
| Flowformer | 26.1 | - | - |
| CCMR+ | 36.0 | - | - |
| WAFT-Twins-a2 | 7.0 | 7.6 | 9.2 |

Table 1: We profile the training memory cost with batch size 1 on an RTX A6000. Our warping method significantly reduces the cost.

128 for the 1/2, 1/4, and 1/8 resolution variants, respectively, to make their computational cost similar (around 350GMACs). As shown in Table 1, the partial cost volume with look-up radius $r = 4$ in SEA-RAFT runs out of memory at 1/2 resolution. WAFT removes the reliance on cost volume, and therefore consumes significantly lower memory than existing methods (Wang et al., 2024; Huang et al., 2022; Jahedi et al., 2024).

**Error from Low Resolution Indexing** Since cost volumes are restricted to low resolution, the predicted flow field must be downsampled for cost volume look-up, which inevitably introduces

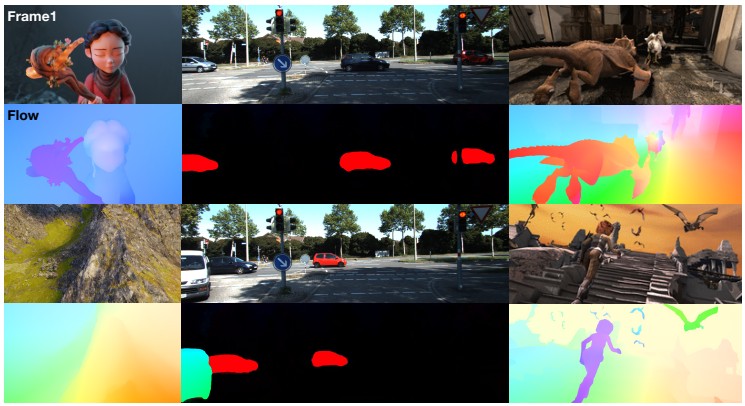

Figure 4: Visualizations on Spring, KITTI, and Sintel public benchmarks (from left to right).

errors. As illustrated in Figure 3 using an example from the Spring benchmark (Mehl et al., 2023), existing methods struggle to produce clear boundaries, particularly noticeable in the top-right corner. In contrast, our method benefits from high-resolution look-up of feature vectors and obtains sharper boundary predictions in these challenging regions. We also show the quantitative performance gain from high-resolution indexing in Table 5.

### 3.3 WARPING VS. COST VOLUME

Warping was widely used in both classical and early deep learning approaches (Memin & Perez, 1998; Brox et al., 2004; Ilg et al., 2017; Ranjan & Black, 2017). However, most recent iterative methods (Wang et al., 2024; Teed & Deng, 2020; Huang et al., 2022) have either replaced warping with cost volumes or used both in combination, since cost volumes have been shown to remarkably improve the performance (Sun et al., 2018a). In this section, we analyze the similarities and differences between warping and cost volume (see Figure 2), and argue that with appropriate designs, warping-based methods can achieve performance on par with cost-volume-based methods.

The overlap of warping and cost volumes lies in their use of the current flow prediction to index into feature maps, which is closely related to optimization (Brox et al., 2004). In cost-volume-based iterative methods, the current flow estimate $f_{cur}$ is used to define partial cost volume $V_{par}$ (see Section 3.1). It is also used to define the warped feature map $\texttt{Warp}(f_{\text{cur}}) \in \mathbb{R}^{h \times w \times d}$, where the feature vector of pixel $p \in I_1$ is indexed from the feature map of frame 2, formulated as:

$$\texttt{Warp}(f_{\text{cur}})_p = F(I_2)_{p+(f_{\text{cur}})_p}$$

Compared to cost volumes, for each pixel in frame 1, warping does not calculate its visual similarity to multiple pixels in frame 2, making it no longer able to explicitly model large displacements. However, we can implicitly handle this long-range dependence through the attention mechanism in vision transformers (Dosovitskiy et al., 2020; Ranftl et al., 2021), which, as we will demonstrate, is crucial to make warping work well (see Section 5.5). The high memory efficiency of warping also enables high-resolution indexing, leading to improved accuracy.

## 4 METHOD

In this section, we describe WAFT, our warping-based iterative method, shown in Figure 1. Its design can be understood as a simple meta-architecture that integrates an input encoder and a recurrent update module. We will also discuss the advantages of our design, especially on its strong performance and the simplifications over past designs.

**Input Encoder** We develop two ways to adapt large-scale pre-trained models. Adaptation 1 (a1) is our initial design specific to DAv2. We freeze the entire DAv2, incorporate features from its DPT head, and further refine these features using a ResNet18. Adaptation 2 (a2) is our improved design that works better and supports more backbones, where we only freeze the ViT/CNN backbones. We

make the DPT head trainable and side-tune the features with 3 ResNet blocks. For completeness, we report the results of both.

**Recurrent Update Module**    Similar to existing iterative methods (Teed & Deng, 2020; Wang et al., 2024; Huang et al., 2022), our recurrent update module $R$ iteratively predicts the residual flow updates. At step $t$, we concatenate $F(I_1)$ (feature of frame 1), $\mathrm{Warp}(f_{\mathrm{cur}})$ (warped feature of frame 2, see Section 3.3), and the current hidden state $\mathrm{Hidden}_t \in \mathbb{R}^{h \times w \times d}$ as input. We use a slightly modified DPT (Ranftl et al., 2021) as the architecture of the module.

**Prediction Head & Loss**    We adopt the Mixture-of-Laplace (MoL) loss used in SEA-RAFT (Wang et al., 2024). At step $t$, the hidden state $\mathrm{Hidden}_t$ is used to predict the MoL parameters $M \in \mathbb{R}^{h \times w \times 6}$. They are upsampled to the original image resolution through convex upsampling (Teed & Deng, 2020; Wang et al., 2024).

**Simplifications over Existing Iterative Methods**    We replace cost volumes, which are standard in existing iterative methods (Sun et al., 2018a; Teed & Deng, 2020; Huang et al., 2022; Wang et al., 2024), with high-resolution warping, which is more memory-efficient. In addition, we have removed the context encoder (Sun et al., 2018a), another flow-specific design standard in existing iterative methods.

A direct benefit from our simplified design is that we can load pre-trained weights for standard architectures such as ViT (Dosovitskiy et al., 2020), which can improve generalization as our experiments will show Section 5.5.

The simplicity of our meta-architecture also enables more apples-to-apples comparisons between direct methods and iterative methods. Existing direct methods (Weinzaepfel et al., 2022; 2023; Saxena et al., 2024) share an input format similar to that of the first iteration of WAFT, making them more directly compatible. We will empirically show the effectiveness and necessity of iterative indexing within our meta-architecture in Section 5.5.

## 5 EXPERIMENTS

We report results on Sintel (Butler et al., 2012), KITTI (Geiger et al., 2013), and Spring (Mehl et al., 2023). Following existing work (Teed & Deng, 2020; Huang et al., 2022; Wang et al., 2024; Morimitsu et al., 2025), for training, we use FlythingChairs (Dosovitskiy et al., 2015), FlyingThings (Mayer et al., 2016), HD1K (Kondermann et al., 2016), Sintel (Butler et al., 2012), KITTI (Geiger et al., 2013), Spring (Mehl et al., 2023), and TartanAir (Wang et al., 2020). We use the widely adopted metrics: endpoint-error (EPE), 1-pixel outlier rate (1px), percentage of flow outliers (Fl), and weighted area under the curve (WAUC). Definitions can be found in (Richter et al., 2017; Mehl et al., 2023; Geiger et al., 2013; Morimitsu et al., 2025).

### 5.1 ARCHITECTURE DETAILS

**Input Encoder**    We use frozen ImageNet-pretrained Twins-SVT-Large (Chu et al., 2021), depth-pretrained DAv2-S (Yang et al., 2024), and unsupervised-pretrained DINOv3-ViT-S (Siméoni et al., 2025) in input encoders.

**Recurrent Update Module**    We use a modified DPT-Small (Ranftl et al., 2021) as the recurrent update module. We concatenate the image features and use a $1 \times 1$ conv to obtain the initial hidden state. Since the image features are already $2\times$ downsampled, we change the patch size to 8. We set the resolution of the positional embedding to $224 \times 224$, and interpolate it for other resolutions. We use $T = 5$ iterations in training and inference.

| Method | 1px↓ | EPE↓ | Fl↓ | WAUC↑ |
|---|---|---|---|---|
| FlowNet2 (Ilg et al., 2017)* | 6.710 | 1.040 | 2.823 | 90.907 |
| SpyNet (Ranjan & Black, 2017)* | 29.963 | 4.162 | 12.866 | 67.150 |
| PWC-Net (Sun et al., 2018a)* | 82.27 | 2.288 | 4.889 | 45.670 |
| RAFT (Teed & Deng, 2020)* | 6.790 | 1.476 | 3.198 | 90.920 |
| GMA (Jiang et al., 2021)* | 7.074 | 0.914 | 3.079 | 90.722 |
| FlowFormer (Huang et al., 2022)* | 6.510 | 0.723 | 2.384 | 91.679 |
| GMFlow (Xu et al., 2022)* | 10.355 | 0.945 | 2.952 | 82.337 |
| RPKNet (Morimitsu et al., 2024) | 4.809 | 0.657 | 1.756 | 92.638 |
| CroCoFlow (Weinzaepfel et al., 2023) | 4.565 | 0.498 | 1.508 | 93.660 |
| Win-Win (Leroy et al., 2023) | 5.371 | 0.475 | 1.621 | 92.270 |
| SEA-RAFT(M) (Wang et al., 2024) | 3.686 | 0.363 | 1.347 | 94.534 |
| DPFlow (Morimitsu et al., 2025) | 3.442 | 0.340 | 1.311 | 94.980 |
| **WAFT-DAv2-a1-540p** | **3.418** | 0.340 | **1.280** | 94.663 |
| **WAFT-DAv2-a1-1080p** | **3.347** | 0.337 | **1.222** | **95.189** |
| **WAFT-Twins-a2** | **3.268** | 0.331 | 1.282 | 94.786 |
| **WAFT-DAv2-a2** | **3.298** | **0.304** | **1.197** | **94.990** |
| **WAFT-DINOv3-a2** | **3.182** | 0.325 | 1.246 | **95.051** |

Table 3: WAFT ranks 1st on Spring (Mehl et al., 2023) on all metrics. We highlight all SOTA performance. * denotes the submissions from the Spring team.

| Type | Method | Sintel | | KITTI | | Inference Cost | |
|---|---|---|---|---|---|---|---|
| | | Clean↓ | Final↓ | All↓ | Non-Occ↓ | #MACs (G) | Latency (ms) |
| Direct | GMFlow (Xu et al., 2022) | 1.74 | 2.90 | 9.32 | 3.80 | 603 | 139 |
| | CroCoFlow (Weinzaepfel et al., 2023) | 1.09 | 2.44 | 3.64 | 2.40 | 57343 | 6422 |
| | DDVM (Saxena et al., 2024) | 1.75 | 2.48 | **3.26** | 2.24 | - | - |
| Iterative w/ Cost Volume | PWC-Net+ (Sun et al., 2019) | 3.45 | 4.60 | 7.72 | 4.91 | 101 | 24 |
| | RAFT (Teed & Deng, 2020) | 1.61 | 2.86 | 5.10 | 3.07 | 938 | 141 |
| | DIP (Zheng et al., 2022) | 1.44 | 2.83 | 4.21 | 2.43 | 3068 | 499 |
| | GMFlowNet (Zhao et al., 2022) | 1.39 | 2.65 | 4.79 | 2.75 | 1094 | 244 |
| | CRAFT (Sui et al., 2022) | 1.45 | 2.42 | 4.79 | 3.02 | 2274 | 483 |
| | FlowFormer (Huang et al., 2022) | 1.20 | 2.12 | 4.68 | 2.69 | 1715 | 336 |
| | GMFlow+ (Xu et al., 2023b) | 1.03 | 2.37 | 4.49 | 2.40 | 1177 | 250 |
| | RPKNet (Morimitsu et al., 2024) | 1.31 | 2.65 | 4.64 | 2.71 | 137 | 183 |
| | CCMR+ (Jahedi et al., 2024) | 1.07 | 2.10 | 3.86 | 2.07 | 12653 | 999 |
| | MatchFlow(G) (Dong et al., 2023) | 1.16 | 2.37 | 4.63 | 2.77 | 1669 | 291 |
| | Flowformer++(Shi et al., 2023) | 1.07 | **1.94** | 4.52 | - | 1713 | 374 |
| | SEA-RAFT(L) (Wang et al., 2024) | 1.31 | 2.60 | 4.30 | - | 655 | 108 |
| | AnyFlow (Jung et al., 2023) | 1.23 | 2.44 | 4.41 | 2.69 | - | - |
| | FlowDiffuser (Luo et al., 2024) | 1.02 | 2.03 | 4.17 | 2.82 | 2466 | 599 |
| | SAMFlow (Zhou et al., 2024) | 1.00 | 2.08 | 4.49 | - | 9717 | 1757 |
| | DPFlow (Morimitsu et al., 2025) | 1.04 | 1.97 | 3.56 | 2.12 | 414 | 131 |
| Iterative w/ Warping | SpyNet (Ranjan & Black, 2017) | 6.64 | 8.36 | 35.07 | 26.71 | 167 | 25 |
| | FlowNet2 (Ilg et al., 2017) | 4.16 | 5.74 | 10.41 | 6.94 | 230 | 75 |
| | **WAFT-DAv2-a1** | 1.09 | 2.34 | 3.42 | **2.04** | 853 | 240 |
| | **WAFT-Twins-a2** | 1.02 | 2.39 | 3.53 | 2.12 | 1020 | 290 |
| | **WAFT-DAv2-a2** | **0.95** | 2.33 | 3.31 | **2.03** | 807 | 240 |
| | **WAFT-DINOv3-a2** | **0.94** | 2.02 | 3.56 | 2.13 | 732 | 212 |

Table 2: We report endpoint-error (EPE) on Sintel (Butler et al., 2012), Fl on KITTI (Geiger et al., 2013), and highlight all SOTA performance. On KITTI, WAFT ranks first on non-occluded pixels and second on all pixels. It also achieves state-of-the-art performance on Sintel (clean). We measure the latency on an RTX3090 with batch size 1 and 540p input.

## 5.2 TRAINING DETAILS

**Benchmark Submissions** Following SEA-RAFT (Wang et al., 2024), we first pre-train our model on TartanAir (Wang et al., 2020) for 300k steps, with a batch size of 32 and learning rate $4 \times 10^{-4}$. We fine-tune our model on FlyingChairs (Zhao et al., 2020a) with the same hyperparameters for 50k steps, and then fine-tune it on FlyingThings (Mayer et al., 2016) for 200k steps. For all submissions, we keep the batch size as 32 by default and reduce the learning rate to $10^{-4}$. For KITTI (Geiger et al., 2013) submission, we fine-tune our model on KITTI(train) for 5k steps. For Sintel (Butler et al., 2012) submission, we follow previous work to fine-tune our model on the mixture of FlyingThings, HD1K (Kondermann et al., 2016), KITTI(train), and Sintel(train) for 200k steps. For Spring (Mehl et al., 2023) submissions, we fine-tune our models on Spring(train) for 200k steps with a batch size of 32. We train an extra 1080p WAFT-DAv2-a1 model with a batch size of 8.

**Zero-Shot Evaluation** We first train our model on FlyingChairs for 50k steps, and then fine-tune it on FlyingThings for 50k steps. The batch size is set to 32, and the learning rate is set to $10^{-4}$.

## 5.3 BENCHMARK RESULTS

**Sintel & KITTI** Results are shown in Table 2. Using a Twins backbone only pre-trained on ImageNet, WAFT ranks second on KITTI and is competitive on Sintel (clean). It outperforms prior cost-volume-based SOTA Flowformer++ (Shi et al., 2023) in both accuracy and efficiency given the same backbone, demonstrating the strength of high-resolution warping. The performance can be further improved with stronger backbones (Yang et al., 2024; Siméoni et al., 2025). Using a depth-pretrained DAv2 (Yang et al., 2024), on KITTI (Geiger et al., 2013), WAFT achieves the best Fl on non-occluded pixels and the second best on all pixels. It also ranks first on Sintel (Clean) (Butler et al., 2012) and competitive on Sintel (Final). WAFT is 1.3-4.1× faster than existing methods that have competitive accuracy (e.g., 1.3× than Flowformer++ Shi et al. (2023), 4.1× than CCMR+ Jahedi et al. (2024)), demonstrating its high efficiency.

Note that there is an outlier sequence, 'Ambush 1', which severely affects the average performance on Sintel (Final) as mentioned in Saxena et al. (2024). We show that WAFT outperforms Flow-

former++ on Sintel (Final) with the same Twins backbone when 'Ambush 1' is excluded. More details can be found in Table 6.

**Spring**  Results are shown in Table 3. Following the downsample-upsample protocol (540p) of SEA-RAFT (Wang et al., 2024), WAFT outperforms existing methods on EPE and 1px with a Twins backbone only pre-trained on ImageNet. We also show that WAFT achieves the best performance on all metrics with a depth-pretrained DAv2 backbone. Benefiting from warping, WAFT can be trained at full resolution (1080p) to improve the performance further.

**Comparison with Existing Warping-based Methods**  It appears that warping as a network operation has been largely abandoned by works in the last 8 years and the last time warping-based methods achieved top positions on the leaderboards were around 2017 (Ranjan & Black, 2017; Ilg et al., 2017). WAFT is significant in that it has revisited and revived an idea that has fallen out of favor. Compared to methods that do use warping (Ranjan & Black, 2017; Ilg et al., 2017), WAFT reduces endpoint-error (EPE) by at least 64% on Sintel and 70% on Spring (Mehl et al., 2023), while also reducing Fl by at least 68% on KITTI (Geiger et al., 2013) and 57% on Spring (Mehl et al., 2023). Besides, WAFT reduces 1px-outlier rate by 52% on Spring (Mehl et al., 2023).

## 5.4 ZERO-SHOT EVALUATION

Following previous work (Teed & Deng, 2020; Huang et al., 2022; Sun et al., 2018a), we train our model on FlyingChairs (Dosovitskiy et al., 2015) and FlyingThings (Mayer et al., 2016). Then we evaluate the performance on the training split of Sintel (Butler et al., 2012) and KITTI (Geiger et al., 2013).

**Analysis**  Results are shown in Table 4. WAFT achieves strong cross-dataset generalization. On KITTI (train), WAFT outperforms other methods by a large margin with an ImageNet-pretrained Twins backbone: It improves the endpoint-error (EPE) from 3.37 to 2.98 and Fl from 11.1 to 9.9. On Sintel (train), WAFT achieves performance close to state-of-the-art methods. Compared to the previous warping-based method (Ilg et al., 2017), WAFT improves the performance by at least 31%.

| Method | Sintel (train) | | KITTI (train) | |
|---|---|---|---|---|
| | Clean↓ | Final↓ | Fl-epe↓ | Fl-all↓ |
| PWC-Net (Sun et al., 2018a) | 2.55 | 3.93 | 10.4 | 33.7 |
| RAFT (Teed & Deng, 2020) | 1.43 | 2.71 | 5.04 | 17.4 |
| GMA (Jiang et al., 2021) | 1.30 | 2.74 | 4.69 | 17.1 |
| SKFlow (Sun et al., 2022) | 1.22 | 2.46 | 4.27 | 15.5 |
| DIP (Zheng et al., 2022) | 1.30 | 2.82 | 4.29 | 13.7 |
| EMD-L (Deng et al., 2023) | 0.88 | 2.55 | 4.12 | 13.5 |
| CRAFT (Sui et al., 2022) | 1.27 | 2.79 | 4.88 | 17.5 |
| RPKNet (Morimitsu et al., 2024) | 1.12 | 2.45 | - | 13.0 |
| GMFlowNet (Zhao et al., 2022) | 1.14 | 2.71 | 4.24 | 15.4 |
| FlowFormer (Huang et al., 2022) | 1.01 | 2.40 | 4.09 | 14.7 |
| Flowformer++ (Shi et al., 2023) | 0.90 | 2.30 | 3.93 | 14.2 |
| CCMR+ (Jahedi et al., 2024) | 0.98 | 2.36 | - | 12.9 |
| MatchFlow(G) (Dong et al., 2023) | 1.03 | 2.45 | 4.08 | 15.6 |
| SEA-RAFT(L) (Wang et al., 2024) | 1.19 | 4.11 | 3.62 | 12.9 |
| AnyFlow (Jung et al., 2023) | 1.10 | 2.52 | 3.76 | 12.4 |
| SAMFlow (Zhou et al., 2024) | 0.87 | **2.11** | 3.44 | 12.3 |
| FlowDiffuser (Luo et al., 2024) | **0.86** | 2.19 | 3.61 | 11.8 |
| DPFlow (Morimitsu et al., 2025) | 1.02 | 2.26 | 3.37 | 11.1 |
| FlowNet2 (Ilg et al., 2017) | 2.02 | 3.14 | 10.1 | 30.4 |
| **WAFT-DAv2-a1** | 1.00 | 2.15 | **3.10** | **10.3** |
| **WAFT-Twins-a2** | 1.02 | 2.46 | **2.98** | **9.9** |
| **WAFT-DAv2-a2** | 1.01 | 2.49 | **3.28** | **10.9** |
| **WAFT-DINOv3-a2** | 1.28 | 2.56 | 3.49 | 12.9 |

Table 4: WAFT achieves the best cross-dataset generalization on KITTI(train), reducing the error by 11%. We highlight all SOTA performance.

## 5.5 ABLATION STUDY

We conduct zero-shot ablations in Table 5 on the training split of Sintel (Butler et al., 2012) and the sub-val split (Wang et al., 2024) of Spring (Mehl et al., 2023) based on WAFT-DAv2-a1. In all experiments, the models are trained on FlythingThings (Mayer et al., 2016) for 50k steps, with a batch size of 32 and learning rate $10^{-4}$. The average EPE and 1px are reported.

**Different Input Encoder**  Both pre-trained weights and adaptations are important to the performance. Note that the strong performance of WAFT is not merely from advanced backbones. Using a Twins backbone only pre-trained on ImageNet as adopted in Flowformer (Huang et al., 2022), WAFT ranks first on Spring, second on KITTI, and is competitive on Sintel. More details can be found in Table 2, 3, and 4.

**Different Recurrent Update Module**  The vision transformer design is crucial to iterative warping. We observe a significant performance drop when replacing the DPT-based recurrent update module with CNNs, highlighting the importance of modeling long-range dependence. This finding may help explain why early deep learning approaches (Ilg et al., 2017; Ranjan & Black, 2017) that

| Experiment | Input Enc. | Rec. Upd. | #Steps | Index Reso. | Sintel(train) | | Spring(sub-val) | | #MACs |
|---|---|---|---|---|---|---|---|---|---|
| | | | | | Clean↓ | Final↓ | EPE↓ | 1px↓ | |
| WAFT-DAv2-a1 | DAv2-S+Res18 | DPT-S | 5 | 1/2 | 1.18 | 2.33 | 0.27 | 1.43 | 858G |
| Different input enc. | Res18 DAv2-S | DPT-S | 5 | 1/2 | 1.27 1.55 | 2.81 2.64 | 0.27 0.37 | 1.59 2.70 | 600G 670G |
| DAv2 w/o pre-train | DAv2-S+Res18 | DPT-S | 5 | 1/2 | 1.42 | 2.74 | 0.28 | 1.77 | 858G |
| Different rec. upd. | DAv2-S+Res18 | Res18 ConvGRU | 5 | 1/2 | 7.23 2.79 | 6.84 4.80 | 0.45 0.39 | 2.93 2.71 | 1098G 800G |
| 1/8 reso. + warp | DAv2-S+Res18 | DPT-S | 5 | 1/8 | 1.15 | 2.31 | 0.32 | 1.82 | 859G |
| 1/8 reso. + corr. | DAv2-S+Res18 | DPT-S | 5 | 1/8 | 1.10 | 2.45 | 0.33 | 1.74 | 883G |
| Direct variants | DAv2-B+Res18 | DPT-S DPT-B | 1 | 1/2 | 2.36 2.37 | 3.43 3.38 | 0.59 0.61 | 10.5 11.1 | 1009G 1277G |
| Image-space warp | DAv2-S+Res18 | DPT-S | 5 | 1/2 | 1.28 | 2.50 | 0.27 | 1.37 | 1902G |
| Refine w/o warp | DAv2-S+Res18 | DPT-S | 5 | 1/2 | 2.04 | 3.42 | 0.58 | 9.44 | 858G |
| w/ Context | DAv2-S+Res18 | DPT-S | 5 | 1/2 | 1.22 | 2.32 | 0.29 | 1.70 | 1005G |

Table 5: We report the zero-shot ablation results on Sintel(train) (Butler et al., 2012) and Spring(sub-val) (Mehl et al., 2023; Wang et al., 2024). The effect of changes can be identified through comparisons with the first row. See Section 5.5 for details.

implemented warping using CNNs underperformed compared to cost-volume-based methods (Sun et al., 2018a; Teed & Deng, 2020; Huang et al., 2022).

**High-Resolution Indexing**  High-resolution indexing into feature maps using current flow estimates remarkably improves performance. We implement variants that index at 1/8 resolution by changing the patch size of DPT to $2 \times 2$, and find that high-resolution indexing significantly improves 1px-outlier rate on Spring(sub-val) (Mehl et al., 2023).

We also design a cost-volume-based variant following the common setup (Wang et al., 2024; Teed & Deng, 2020; Huang et al., 2022) with look-up radius $4$ at 1/8 resolution, and find that it performs similarly to the warping counterpart (shown in Table 5) but costs $2.2\times$ training memory (21.2 GiB vs. 9.5 GiB).

**Direct vs. Iterative**  Iterative updates achieve better performance than direct regression within our meta-architecture. We implement direct regression by setting the number of iterations $T = 1$. For fair comparison, we scale up the networks to match the computational cost of 5-iteration WAFT. Our results show that WAFT significantly outperforms these direct regression variants, indicating the effectiveness and high efficiency of the iterative paradigm. This finding aligns with the observation that existing direct methods either underperform (Xu et al., 2022; Weinzaepfel et al., 2022) or require substantially more computational cost (Saxena et al., 2024; Weinzaepfel et al., 2022; 2023) compared to iterative approaches (Teed & Deng, 2020; Wang et al., 2024; Huang et al., 2022; Morimitsu et al., 2025).

**Warping Features vs. Pixels**  Feature-space warping is more effective than image-space warping, which is commonly used in classic methods (Ma et al., 2022; Brox et al., 2004; Black & Anandan, 1996) and early deep learning methods (Ilg et al., 2017; Ranjan & Black, 2017). Feature-space warping does not need to re-extract features for the warped image in each iteration, significantly saving computational cost while achieving slightly better accuracy.

**Effectiveness of Warping**  It is possible to perform iterative updates without warping the features using the current flow estimates. We can simply use the original feature maps as input to the update module. Compared to this baseline, warping has significantly lower error with a negligible cost. This observation aligns with the conclusions of previous work (Brox et al., 2004), which theoretically justifies the combination of warping and recurrent updates by framing it as a fixed-point iteration algorithm.

**Context Encoder** Prior work (Sun et al., 2018a; Teed & Deng, 2020; Wang et al., 2024; Huang et al., 2022) has often used a context encoder that provides an extra input to the update module. Our ablation show that the context encoder is not necessary. The context encoder introduces additional computation overhead, but does not significantly affect performance. Previous work (Wang et al., 2024) also points out that the context encoder can be regarded as a direct flow regressor, which functions similarly to the first iteration of WAFT.

ACKNOWLEDGMENTS

This work was partially supported by the National Science Foundation.

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

| Sequence | WAFT-Twins-a2 | DPFlow | Flowformer++ | FlowDiffuser | DDVM | SAMFlow |
|---|---|---|---|---|---|---|
| Perturbed Market 3 | 0.893 (0.460) | 0.892 (0.423) | 0.958 (0.511) | 0.897 (0.490) | 0.787 (0.372) | 0.932 (0.493) |
| Perturbed Shaman 1 | 0.174 (0.163) | 0.213 (0.201) | 0.251 (0.236) | 0.267 (0.255) | 0.219 (0.196) | 0.241 (0.219) |
| Ambush 1 | 20.561 (2.526) | 8.366 (2.970) | 6.610 (2.605) | 7.224 (2.583) | 29.33 (14.07) | 10.586 (2.733) |
| Ambush 3 | 3.173 (2.048) | 3.019 (1.729) | 2.939 (1.816) | 3.148 (1.828) | 2.855 (3.016) | 3.411 (1.779) |
| Bamboo 3 | 0.460 (0.423) | 0.486 (0.438) | 0.546 (0.513) | 0.594 (0.508) | 0.415 (0.380) | 0.522 (0.473) |
| Cave 3 | 2.199 (1.567) | 2.341 (1.631) | 2.344 (1.477) | 2.464 (1.433) | 2.042 (1.658) | 2.475 (1.445) |
| Market 1 | 0.851 (0.384) | 0.890 (0.491) | 1.073 (0.550) | 1.238 (0.517) | 0.719 (0.467) | 1.060 (0.491) |
| Market 4 | 7.513 (3.933) | 7.939 (3.834) | 8.086 (4.450) | 8.024 (4.027) | 5.517 (3.971) | 6.636 (3.680) |
| Mountain 2 | 0.366 (0.087) | 0.177 (0.078) | 0.288 (0.118) | 0.409 (0.101) | 0.176 (0.095) | 0.500 (0.217) |
| Temple 1 | 0.511 (0.297) | 0.511 (0.302) | 0.657 (0.359) | 0.789 (0.340) | 0.452 (0.284) | 0.853 (0.340) |
| Tiger | 0.463 (0.333) | 0.571 (0.411) | 0.595 (0.430) | 0.636 (0.391) | 0.413 (0.344) | 0.573 (0.392) |
| Wall | 1.708 (0.979) | 2.011 (1.231) | 1.727 (0.793) | 1.692 (0.734) | 1.639 (3.210) | 2.105 (0.780) |
| Avg | 2.393 (1.015) | 1.975 (1.046) | 1.943 (1.073) | 2.026 (1.016) | 2.475 (1.754) | 0.995 (2.080) |
| Avg (w/o Ambush 1) | 1.639 (0.952) | 1.710 (0.966) | 1.750 (1.010) | 1.810 (0.951) | 1.360 (1.242) | 1.727 (0.923) |

Table 6: We report the endpoint-error (EPE) on all sequences of Sintel (Butler et al., 2012), shown in the format "final-epe (clean-epe)". We highlight the best result on each sequence.

Lihe Yang, Bingyi Kang, Zilong Huang, Zhen Zhao, Xiaogang Xu, Jiashi Feng, and Hengshuang Zhao. Depth anything v2. *Advances in Neural Information Processing Systems*, 37:21875–21911, 2024.

Christopher Zach, Thomas Pock, and Horst Bischof. A duality based approach for realtime tv-l 1 optical flow. In *Pattern Recognition: 29th DAGM Symposium, Heidelberg, Germany, September 12-14, 2007. Proceedings 29*, pp. 214–223. Springer, 2007.

Shengyu Zhao, Yilun Sheng, Yue Dong, Eric I Chang, Yan Xu, et al. Maskflownet: Asymmetric feature matching with learnable occlusion mask. In *Proceedings of the IEEE/CVF Conference on Computer Vision and Pattern Recognition*, pp. 6278–6287, 2020a.

Shiyu Zhao, Long Zhao, Zhixing Zhang, Enyu Zhou, and Dimitris Metaxas. Global matching with overlapping attention for optical flow estimation. In *Proceedings of the IEEE/CVF Conference on Computer Vision and Pattern Recognition*, pp. 17592–17601, 2022.

Yang Zhao, Gangwei Xu, and Gang Wu. Hybrid cost volume for memory-efficient optical flow. In *Proceedings of the 32nd ACM International Conference on Multimedia*, pp. 8740–8749, 2024.

Yuxuan Zhao, Ka Lok Man, Jeremy Smith, Kamran Siddique, and Sheng-Uei Guan. Improved two-stream model for human action recognition. *EURASIP Journal on Image and Video Processing*, 2020(1):1–9, 2020b.

Zihua Zheng, Ni Nie, Zhi Ling, Pengfei Xiong, Jiangyu Liu, Hao Wang, and Jiankun Li. Dip: Deep inverse patchmatch for high-resolution optical flow. In *Proceedings of the IEEE/CVF Conference on Computer Vision and Pattern Recognition*, pp. 8925–8934, 2022.

Shili Zhou, Ruian He, Weimin Tan, and Bo Yan. Samflow: Eliminating any fragmentation in optical flow with segment anything model. In *Proceedings of the AAAI Conference on Artificial Intelligence*, volume 38, pp. 7695–7703, 2024.

Yiming Zuo and Jia Deng. View synthesis with sculpted neural points. *arXiv preprint arXiv:2205.05869*, 2022.

# A  APPENDIX

**Sintel Results**  We show the sequence-wise results of several representative methods (Morimitsu et al., 2025; Huang et al., 2022; Luo et al., 2024; Saxena et al., 2024; Zhou et al., 2024) on Sintel in Table 6. The sequence 'Ambush 1' appears to be an outlier which severely affects the average EPE on the final split. Given the same ImageNet-pretrained Twins backbone, WAFT outperforms SOTA Flowformer++ when 'Ambush 1' is excluded.

