# OpenReview forum: "WAFT: Warping-Alone Field Transforms for Optical Flow"
_ICLR.cc/2026/Conference — ICLR 2026 Oral_

### Official Review · Reviewer_abBA · 2025-10-26

**Soundness:** 3
**Presentation:** 3
**Contribution:** 3
**Rating:** 6
**Confidence:** 4

**Summary:**

This paper proposes a simplified meta-architecture for optical flow estimation without using cost volumes. The proposed WAFT algorithm consists of an input encoder which leverages existing large-scale pre-trained models for feature extraction, and a recurrent update module based on vision transformers that can iteratively updates optical flow with large displacements. Experiment shows that the proposed WAFT algorithm achieves top rankings on various benchmarks, including Spring and KITTI, furthermore, it does so with significantly lower memory cost and up to 4.1x faster inference times.

**Strengths:**

1. The design of WAFT without cost volume computation is very simple, flexible and effective, making it a significant contribution for computer vision research community
2. By avoiding cost volumes computation, WAFT can perform warping on original resolution feature maps, which can help achieving sharper boundary predictions in optical flow estimation
3. WAFT has shown best zero-shot cross-dataset generalization on KITTI, which is an important property towards generalization capability on unseen data.
4. The paper is well-structured and clearly written.

**Weaknesses:**

1. The iterative recurrent update module may restrict the algorithm's potential for parallel optimization to achieve low latency.
2. WAFT relies on existing pre-trained vision foundation models, which may limits its potential for further computational efficiency improvement on feature extraction.
3. Compared with improved memory and computational efficiency improvement, the improvement on flow accuracy is relatively limited.

**Questions:**

1. In table 2, while the ratio of MACs reduction is high (CCMR+'s 12653 vs. WAFT-DAv2-a1's 853), the speed up of latency is not at the same scale (CCMR+'s 999 vs. WAFT-DAv2-a1's 240), it would be good to give further explanation on this.

---

> ### Author Response · Authors · 2025-11-21
>
> We thank the reviewer for the detailed feedback. In the following, we address the concerns.
>
> Q1: "The iterative recurrent update module may restrict the algorithm's potential for parallel optimization to achieve low latency."
>
> A1: The use of iterative updates by itself does not necessarily mean worse latency, when compared to “direct” methods that do not use iterative updates. This is because a direct method can still have many sequential layers within its “one-step” image encoder, which results in high latency, whereas an iterative method can have many light-weight iterations and low latency overall. Thus it is in principle possible for WAFT to be parallel and fast within each iteration and have better overall latency. In addition, our existing experiments show that WAFT outperforms all existing direct methods (those that do not use iterative updates) in both accuracy and latency (see Tables 1 and 2).
>
> Q2: "WAFT relies on existing pre-trained vision foundation models, which may limit its potential for further computational efficiency improvement on feature extraction."
>
> A2: Adaptation from pre-trained models does not limit the potential for further efficiency improvement, because WAFT can be paired with smaller backbones optimized for efficiency.  In fact, WAFT can achieve state-of-the-art performance with a feature extractor only pre-trained on ImageNet (WAFT-Twins-a2). It is relatively easy to pre-train small models on ImageNet. Also, there are already many efficient, open-source ImageNet-pretrained models [1,2].
>
> Q3: "Compared with improved memory and computational efficiency improvement, the improvement on flow accuracy is relatively limited."
>
> A3: We acknowledge the accuracy improvement on benchmark submissions are not as big as efficiency improvement. However, efficiency is very important for applications, because one can usually improve accuracy by training bigger models, but such models can be too expensive for applications.
>
> In addition, WAFT has a large accuracy improvement on generalization; it reduced the cross-dataset generalization error by 11\% (Table 4). WAFT also revives the warping approach, which has been largely abandoned for the past 8 years. Compared with previous warping-based methods, WAFT reduces the error by at least 31%, up to 68\% (Line 382-389, 408-410).
>
> Q4: "In table 2, while the ratio of MACs reduction is high (CCMR+'s 12653 vs. WAFT-DAv2-a1's 853), the speed up of latency is not at the same scale (CCMR+'s 999 vs. WAFT-DAv2-a1's 240), it would be good to give further explanation on this."
>
> A4: One possible reason is the high parallel compute capability of GPUs. On GPUs, the reduction in MACs typically does not translate proportionally to latency improvements. For instance, increasing the batch size from 1 to 2 doubles the MACs but only slightly increases latency, as many operations run in parallel. MACs is therefore a more “theoretical” measure of computational cost, and it correlates more closely with latency on edge platforms with limited parallel compute capability.
>
> [1] Qin, Danfeng, et al. "MobileNetV4: Universal models for the mobile ecosystem."  *ECCV2024*
>
> [2] Tan, Mingxing, and Quoc Le. "Efficientnet: Rethinking model scaling for convolutional neural networks." PMLR 2019

---

### Official Review · Reviewer_oRyW · 2025-10-31

**Soundness:** 4
**Presentation:** 3
**Contribution:** 4
**Rating:** 8
**Confidence:** 5

**Summary:**

This paper points out the drawback of constructing cost volume in the optical flow field, and propose to replace the cost volume with warping. To achieve competitive performance, the authors propose to utilize: 1) stronger feature encoder 2) high-resolution warping 3) attention- (and pretrained) based updater.

**Strengths:**

1. This paper is well written.
2. This paper has a clear and extensive ablation study to show the effectiveness of each design choice.
3. The author introduced an attention-based updater to replace the cost volume for feature similarity computation. This design is resonable and novel.

**Weaknesses:**

1. Since the author replaces the commonly used CNN updater with attention-based one, it is better to provides more details of the layers.
2. For the models used in Table 2, what is the downsampled ratio? And which line corresponds to the statement in the abstract "while being up to 4.1× faster than methods with similar performance". From my understanding, WAFT-Twins-a2 uses the same feature encoder as FlowFormer++, achieves similar performance but not significant speedup?
3. Can the authors provide more explaination about why context encoder is not useful in the WAFT architecture?

**Questions:**

Please refer to the Weaknesses section.

**Details Of Ethics Concerns:**

There are no ethics concerns for me.

---

> ### Author Response · Authors · 2025-11-21
>
> We thank the reviewer for the detailed feedback. In the following, we address the concerns.
>
> Q1: "Since the author replaces the commonly used CNN updater with attention-based one, it is better to provide more details of the layers."
>
> A1: The design of the recurrent update module is the same as DPT-Small[1] except the patchifier (See Line 311-319). It has 12 attention layers (from ViT-S), 25.6M parameters in total.
>
> Q2: "For the models used in Table 2, what is the downsampled ratio?"
>
> A2: The downsample ratio is 2x, which is higher than the commonly used 8x ratio in previous work and significantly improves the performance (See Table 5).
>
> Q3: "Which line corresponds to the statement in the abstract "while being up to 4.1× faster than methods with similar performance". From my understanding, WAFT-Twins-a2 uses the same feature encoder as FlowFormer++, achieves similar performance but not significant speedup?"
>
> A3: “up to 4.1x” refers to the best speed-up achieved over CCMR+ (240ms vs. 999ms); it does not mean that we achieve 4.1x over every existing method. . You are correct regarding the comparison between WAFT-Twins-a2 and Flowformer++ on Sintel. We will revise our text to make it more clear.
>
> Q4: "Can the authors provide more explanation about why context encoder is not useful in the WAFT architecture?"
>
> A4: See Line 481-485. We believe that the context encoder functions similarly to the first iteration of WAFT when the flow field is all zeros (no warping) and is therefore, in principle, unnecessary.

---

> > ### Comment · Reviewer_oRyW · 2025-11-28
> > **about the statement in abstract**
> >
> > From my understanding, WAFT-Twins-a2 and Flowformer++ adopts similar feature extractors, achieves similar performance. WAFT-Twins-a2 achieves 1.29x speedup (374s vs 290s). The statement of t,1x speedup is comparison with the slowest method. So the statement in the abstract seems to be over-claimed to me. Could the authors give a revision on this statement? Otherwise, I may lower my rating.

---

> > > ### Author Response · Authors · 2025-11-28
> > >
> > > Yes, we will revise this claim in our final version. We sincerely thank you for the valuable suggestions.

---

> > > ### Author Response · Authors · 2025-12-03
> > > **Revised Draft**
> > >
> > > We have revised all related claims in our paper and have uploaded a revised version:
> > >
> > > Line 17-18, 75-76, 372-373: “WAFT is up to 4.1× faster than methods with similar performance.” $\to$ “WAFT is 1.3-4.1x faster than existing methods that have competitive accuracy (e.g., 1.3x than Flowformer++, 4.1x than CCMR+).”

---

### Official Review · Reviewer_ZeZq · 2025-11-01

**Soundness:** 3
**Presentation:** 2
**Contribution:** 2
**Rating:** 6
**Confidence:** 3

**Summary:**

This paper proposes to abandon the cost volume that's a standard component in deep optical flow architectures, and uses the warped target feature vector alone (plus the source feature vector) for flow estimation. It implicitly builds a global context via self-attention in the recurrent update module, which is why it can get rid of the cost volume.

**Strengths:**

1. Removing the cost volume is a good contribution, which may make estimation of optical flow on high-res images much more feasible.

**Weaknesses:**

1. No detailed evaluation of how the model performs on large displacements, on which WAFT might be slightly weaker than models using a cost volume. The authors could make artificial displacements to stress-test WAFT to see where its limit lies.
2. No details are given for the Recurrent Update Module. For example, how many layers (esp. self attention layers), what's the total param count?

**Questions:**

1. Since no cost volume is adopted, I'm worried that the initial errors may accumulate and become larger as the model iterates. The authors could consider such a perturbation test: in the first iteration, perturb the flow prediction with random values, then see how well the model recovers from it.
2. Another challenging scenario is if there are multiple similar objects (e.g. a table with many cups), how well WASP would perform. Chance is the self attention may overly smooth features across these similar objects and make the prediction more random. Of course this would be challenging for methods **with** a cost volume, but I'm curious if it would be more challenging for WASP.

 (Note: I would not lower my rating if the model is not so robust under such perturbations; this is just to better inform readers whre are the "sweet spots" in which the method performs well.)

---

> ### Author Response · Authors · 2025-11-21
>
> We thank the reviewer for the detailed feedback. In the following, we address the concerns.
>
> Q1: "No detailed evaluation of how the model performs on large displacements, on which WAFT might be slightly weaker than models using a cost volume."
>
> A1: Here we report the results on large displacements (flow distance over 40 pixels) WAFT outperforms existing methods on these metrics, showing that it is also better on large displacements.
>
> | Method        | 1px-s40+$\downarrow$ | EPE-s40+$\downarrow$ | Fl-s40+$\downarrow$ | WAUC-s40+$\uparrow$ |
> | :------------ | :--------------------: | :--------------------: | :-------------------: | :-------------------: |
> | WAFT-Twins-a2 |         18.162         |         2.087         |         5.332         |        79.823        |
> | SEA-RAFT (M)  |         21.237         |         2.371         |         5.642         |        77.013        |
> | Croco-Flow    |         33.134         |         4.046         |         8.250         |        67.534        |
> | Flowformer    |         35.344         |         5.753         |        12.556        |        65.110        |
>
> Q2: "No details are given for the Recurrent Update Module. For example, how many layers (esp. self attention layers), what's the total param count?"
>
> A2: The design of the recurrent update module is the same as DPT-Small[1] except the patchifier (See Line 311-319). It has 12 attention layers (from ViT-S), 25.6M parameters in total.
>
> Q3: "I'm worried that the initial errors may accumulate and become larger as the model iterates. The authors could consider such a perturbation test: in the first iteration, perturb the flow prediction with random values, then see how well the model recovers from it."
>
> A3: Introducing perturbations in the first iteration degrades performance, but WAFT is able to recover and still produce reasonable results. In the table below, we perturb the first-iteration flow prediction by adding Gaussian noise and vary the noise strength by adjusting the standard deviation $\sigma$ ($\sigma=1$ means 1 pixel std. at ½ resolution). We report zero-shot results (4-run average) on KITTI (train) and Sintel (train) under the standard zero-shot setting (See Table 4).
>
> | Model         | Perturbation   | KITTI(train) | Sintel(train) |
> | :------------ | :------------- | :----------: | :-----------: |
> | WAFT-Twins-a2 | $\sigma=0$   |   2.98/9.9   |   1.02/2.46   |
> | WAFT-Twins-a2 | $\sigma=0.5$ |  3.30/10.4  |   1.54/2.92   |
> | WAFT-Twins-a2 | $\sigma=1$   |  3.79/14.3  |   2.20/3.55   |
> | Flowformer    | $\sigma=0$   |  4.09/14.7  |   1.01/2.40   |
> | SEA-RAFT(L)   | $\sigma=0$   |  3.62/12.9  |   1.19/4.11   |
>
> Q4: "Another challenging scenario is if there are multiple similar objects (e.g. a table with many cups), how well WAFT would perform. Of course this would be challenging for methods **with** a cost volume, but I'm curious if it would be more challenging for WAFT."
>
> A4: We selected several scenes containing multiple similar objects from [a publicly available Infinigen dataset](https://github.com/princeton-vl/infinigen/blob/main/docs/PreGeneratedData.md) and found that these scenarios are not more challenging for WAFT than for cost-volume-based methods. Specifically, we examined the visualizations of WAFT-DAv2-a2 and SEA-RAFT (L), both using their Sintel-submission weights. WAFT consistently achieves lower EPE across all these scenes. Here is the [anonymous link](https://anonymous.4open.science/r/anonymous-C9B2/multiple-similar-objects.pdf) to visualizations.
>
> [1] Ranftl, René, Alexey Bochkovskiy, and Vladlen Koltun. "Vision transformers for dense prediction."  *ICCV 2021*
>
> [2] Wang, Yihan, Lahav Lipson, and Jia Deng. "Sea-raft: Simple, efficient, accurate raft for optical flow." *ECCV 2024*
>
> [3] Raistrick, Alexander, et al. "Infinite photorealistic worlds using procedural generation."  *CVPR 2023*

---

> > ### Comment · Reviewer_ZeZq · 2025-11-22
> > **Impressive results**
> >
> > Thanks for providing the extra experiments. They show impressive model robustness. I'd increase my rating to accept.

---

### Meta-Review · Area_Chair_ikeW · 2026-01-05

**Summary:**

This paper presents WAFT, a new optical flow architecture that eliminates the cost volume, a standard component, and instead relies on high-resolution warping and a transformer-based update module.

Reviewers initially raised concerns regarding performance on large displacements, error accumulation, architectural details, and the claimed speedup. The authors' rebuttal provided compelling experimental evidence: WAFT excels on large displacements, shows robustness to flow perturbations, and its generalization is SOTA. They clarified the update module's design and appropriately revised an over-claimed speedup comparison. Reviewers found these responses satisfactory.

While the iterative update may limit parallelization and gains in benchmark accuracy are incremental, WAFT achieves excellent efficiency, memory savings, and superior zero-shot generalization. All major concerns were addressed, leading to a positive consensus.

**Reviewer Concerns:**

See above.

**Reviewer Scores:**

Some reviewers may raise their scores after the rebuttal clarifies and addresses their concerns, while others, who already gave high initial scores, may not.

---

### Decision · Program_Chairs · 2026-01-26

Accept (Oral)